DOI: 10.1038/s41467-017-00238-8　　**OPEN**

# Evolution of new regulatory functions on biophysically realistic fitness landscapes

Tamar Friedlander[1,2], Roshan Prizak[1], Nicholas H. Barton[1] & Gašper Tkačik [1]

Gene expression is controlled by networks of regulatory proteins that interact specifically with external signals and DNA regulatory sequences. These interactions force the network components to co-evolve so as to continually maintain function. Yet, existing models of evolution mostly focus on isolated genetic elements. In contrast, we study the essential process by which regulatory networks grow: the duplication and subsequent specialization of network components. We synthesize a biophysical model of molecular interactions with the evolutionary framework to find the conditions and pathways by which new regulatory functions emerge. We show that specialization of new network components is usually slow, but can be drastically accelerated in the presence of regulatory crosstalk and mutations that promote promiscuous interactions between network components.

---

[1] Institute of Science and Technology Austria, Am Campus 1, A-3400, Klosterneuburg, Austria. [2] Present address: The Robert H. Smith Institute of Plant Sciences and Genetics in Agriculture, Faculty of Agriculture Hebrew University of Jerusalem, P.O. Box 12, Rehovot 7610001, Israel. Tamar Friedlander and Roshan Prizak contributed equally to this work. Correspondence and requests for materials should be addressed to G.T. (email: gtkacik@ist.ac.at)

Phenotypes evolve largely through changes in gene regulation[1–4], and such evolution may be flexible and rapid[5, 6]. Of particular importance are mutations affecting affinity and specificity of transcription factors (TFs) for their upstream signals or for their binding sites, short fragments of DNA that TFs interact with to activate or repress transcription of specific target genes. Mutations in these binding sites or at sites that alter TF specificity are crucial because of their ability to "rewire" the regulatory network—to weaken or completely remove existing interactions and add new ones, either functional or spurious. Emergence of novel functions in such a network will usually be constrained to evolutionary trajectories that maintain a viable pattern of existing interactions. This raises a fundamental question about the effects of such constraints on the accessibility of different regulatory architectures and the timescales needed to reach them.

The case that we focus on here is the divergence of gene regulation, which can give rise to a variety of new phenotypes, e.g., via expansion in TF families. A regulatory function previously accomplished by a single (or several) TF(s) is now carried out by a larger number of TFs, allowing for additional fine-tuning and precision, or, alternatively, for an expansion of the regulatory scope[7–17]. The main avenue for such expansions are gene duplications[18–20]. Rapid weakening of expression of the duplicates[21] or alternatively selection to increase expression[22, 23] facilitate the preservation and fixation of duplicates. Gene duplications generate extra copies of the TFs and thus provide the "raw material" for evolutionary diversification. Subsequent specialization of TFs often involves divergence in both their inputs (e.g., ligands) and outputs (regulated genes)[3, 24]. Examples range from repressors involved in bacterial carbon metabolism that arose from the same ancestor via a series of duplication–divergence events[25], and ancestral TF Lys14 in the metabolism of *Saccharomyces cerevisiae*, which diverged into three different TFs regulating different subsets of genes in *Candida albicans*[26], to many variants of Lim and Pou-homeobox genes involved in neural development across different organisms[27]. In some systems the ligand sensing and gene regulatory functions are distributed across two or more molecules, as for bacterial two-component pathways[28] and eukaryotic signaling cascades[29]; here, too, specialization can occur by a series of mutations in multiple relevant components.

Immediately following a duplication event, molecular recognition between TFs, their input signals, and their binding sites is specific but undifferentiated between the two TF copies. Under selection to specialize, recognition sequences and ligand preferences of the two TFs can diverge, but only if some degree of matching between TFs and their binding sites is continually retained to ensure network function. Binding sites are thus forced to coevolve in tandem with the TFs. Yet little is known about the resulting limits to evolutionary outcomes, the likelihood of potential evolutionary trajectories, and the relevant timescales; specifically, it is unclear how these quantities of interest depend on important parameters, such as the number of regulated genes, the length and specificity of the binding sites, the correlations between the input signals, etc.

Theoretical understanding of TF duplication is still incomplete, with existing models predominantly belonging to two categories. The first category of gene duplication–differentiation models studies subfunctionalization of isolated proteins (e.g., enzymes) that do not have any regulatory role[30]. When cis-regulatory mutations that control the expression of the duplicated gene are included[31–34], this is done in a simplified fashion, e.g., by a small number of discrete alleles that represent TF-binding sites appearing and disappearing at fixed rates[33, 34]. Because this approach ignores the essentials of molecular recognition,

it cannot model co-evolution between TFs and their binding sites—the topic of our interest.

The second category of studies tracks regulatory sequences explicitly and uses a biophysical description of TF–BS (binding site) interactions, properly accounting for the fact that TFs can bind a variety of DNA sequences with different affinities[35–37]. In conjunction with thermodynamic models of gene regulation[38–41], this approach has been used to study the evolution of binding sites given a single TF[37, 42–45], while mostly overlooking the issue of TF duplication and subfunctionalization (but see refs [46, 47]).

Here we synthesize these two frameworks—the biophysical description of gene regulation and the evolutionary modeling of TF specialization—to construct a realistic description of the fundamental step by which regulatory networks have evolved. A biophysical model of this set-up gives rise to complex fitness landscapes that are markedly different from simple forms considered previously; in what follows, we show that realistic landscapes exert a major influence over the evolutionary outcomes and dynamics. The structure of the paper is as follows. We first introduce the basic model with two TFs and two regulated genes, and analyze its steady state distribution of outcomes, showing that the huge genotypic space maps to very few phenotypes. We next analyze the possible dynamical trajectories and timescales leading to each phenotype. Finally, we extend the basic model to a larger number of regulated genes and study the effect of "promiscuity-promoting mutations," i.e., mutations that render TFs less specific for their binding sites.

## Results

**A biophysically realistic fitness landscape.** In our model, $n_{\text{TF}}$ transcription factors regulate $n_G$ genes by binding to sites of length $L$ base pairs; for simplicity, we consider each gene to have one such binding site. The specificity of a TF for any sequence is determined by the TF's preferred (consensus) sequence; sequences matching consensus are assigned lowest energy, $E = 0$, which corresponds to tightest binding, and every mismatch between the consensus and the binding site increases the energy by $\epsilon$; this additive 'mismatch' model has a long history in gene regulation literature[35, 43, 48, 49].

The equilibrium probability that the binding site of gene $j$ ($j = 1,…,n_G$) is bound by active TFs of any type $i$ ($i = 1,…,n_{\text{TF}}$) is a proxy for the gene expression level and is given by the thermodynamic model of gene regulation[38, 50]:

$$p_{jm}\big(\{k_{ij}\}, \{C_i(m)\}\big) = \frac{\sum_i C_i(m) e^{-\epsilon k_{ij}}}{1 + \sum_i C_i(m) e^{-\epsilon k_{ij}}}, \tag{1}$$

where $C_i(m)$ is dimensionless concentration of active TFs of type $i$ in condition $m$, $k_{ij}$ is the number of mismatches between the consensus sequence of the $i$-th TF species and the binding site of the $j$-th gene, and $\epsilon$ is the energy per mismatch in units of $k_B T$. Concentration $C_i(m)$ of active TFs depends on condition $m$, which can represent either time or space (e.g., during developmental gene expression programs) or a discrete external environment (e.g., the presence/absence of particular chemical signals). The simplest case considered here assumes the existence of two such signals that can be either present or absent, in any combination, for a total number of four possible environments ($m = 00, 01, 10, 11$), occurring with probabilities $\alpha_m$; an important parameter will be the correlation, $-1 \leq \rho \leq 1$, between the two signals. Each TF has two binary alleles, $\sigma_i \in [00, 01, 10, 11]$, determining its specificity for the two signals. If the TF $i$ is responsive to a signal and that signal is present in environment $m$, then its active concentration $C_i(m) = C_0$; otherwise, $C_i(m) = 0$. Given constants $C_0$, $\epsilon$, and the genotype $\mathcal{D}$—comprising TF consensus and binding site sequences as well as TF

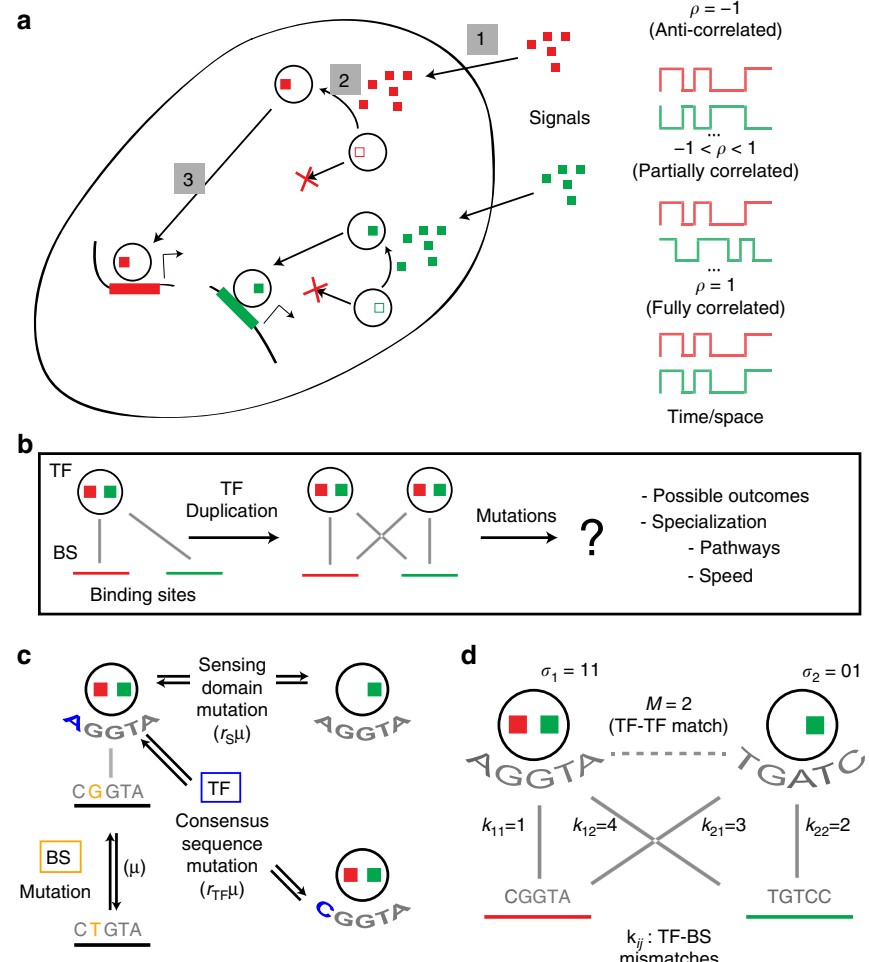

**Fig. 1** Schematic of the model. **a** Simplified physiology of signal transduction: external signaling molecules (*red* and *green squares*) are sensed by the cell (1), activate transcription factors inside the cell (2), which in turn activate the corresponding downstream genes (3). The temporal/spatial appearance of the two external signals can be correlated to different extent, as measured by correlation coefficient, $\rho$. These signals can correspond to different time periods in development, spatial regions in the organism or tissue, or external conditions/ligands. **b** TF, initially responsive to two external signals (*red* and *green* 'slots') and regulating two genes, duplicates and the additional copy fixes in the population. Immediately after duplication, the two copies are undifferentiated. **c** Various mutation types that can occur post-duplication with their associated rates. **d** After accumulating several mutations, the pattern of mismatches between TF consensus sequences and the binding sites is reflected in new values of $\{k_{ij}\}$, which determine the activation levels of the two genes according to Eq. (1). $M$, the number of matches between the consensus sequences of the two TFs (with a value between 0 and $L$), keeps track of the overall divergence of the TF specificities. For a list of model parameters and baseline values see Supplementary Table 1

sensitivity alleles $\sigma_i$—the thermodynamic model of Eq. (1) fully specifies expression levels for all genes in all environments (Supplementary Note 1).

Figure 1a, b illustrates this set-up for a simple case $n_{TF} = n_G = 2$, assuming that the two copies of the TF emerged through an initial gene duplication event and are fixed in the population. The original TF regulates two downstream genes by binding to their binding sites. It is sensitive to both external signals, which can be present with a varying degree of correlation (Fig. 1a). After duplication, three types of mutation can occur, as shown in Fig. 1c: point mutations in the binding sites (rate $\mu$), mutations in the TF coding sequence that change TF's preferred (consensus) specificity (rate $r_{TF}\mu$) and mutations in the two signal-sensing alleles (rate $r_S\mu$), which can give each TF specificity to both signals, to one of them, or to neither. An example in Fig. 1d shows the state of the system after several mutations have affected the degree of (mis)match between the TFs and the binding sites, $k_{ij}$; an especially important quantity that tracks the overall divergence of the TF specificity is denoted as $M$, the match between the two TF consensus sequences.

To complete the evolutionary model, a fitness function is required. We assume selection for the genes to acquire distinct expression patterns in response to external signals, and thus define this fully specialized state as having the highest fitness in our model. Specifically, we penalize the deviations in actual gene expression, $p_{jm}$, from the ideal expression levels, $p_{jm}^*$:

$$F = -s \sum_j \sum_m \alpha_m \beta_{jm} \left( p_{jm} - p_{jm}^* \right)^2, \qquad (2)$$

where the ideal expression level $p_{jm}^*$ is 1 (fully induced) for the first gene if signal 1 is present and the expression is 0 (not induced) otherwise, and similarly for the second gene; $\beta_{jm}$ can be used to vary the relative weight of different errors (e.g. of a gene being uninduced when it should be induced and vice versa, see Supplementary Note 3), and $s$ is the selection intensity. Importantly, selection does not directly depend on the TFs, but only on the expression state of the genes they regulate; genes, however, can only be expressed when TFs bind to proper binding sites, implicitly selecting on TFs. For this reason it is also very

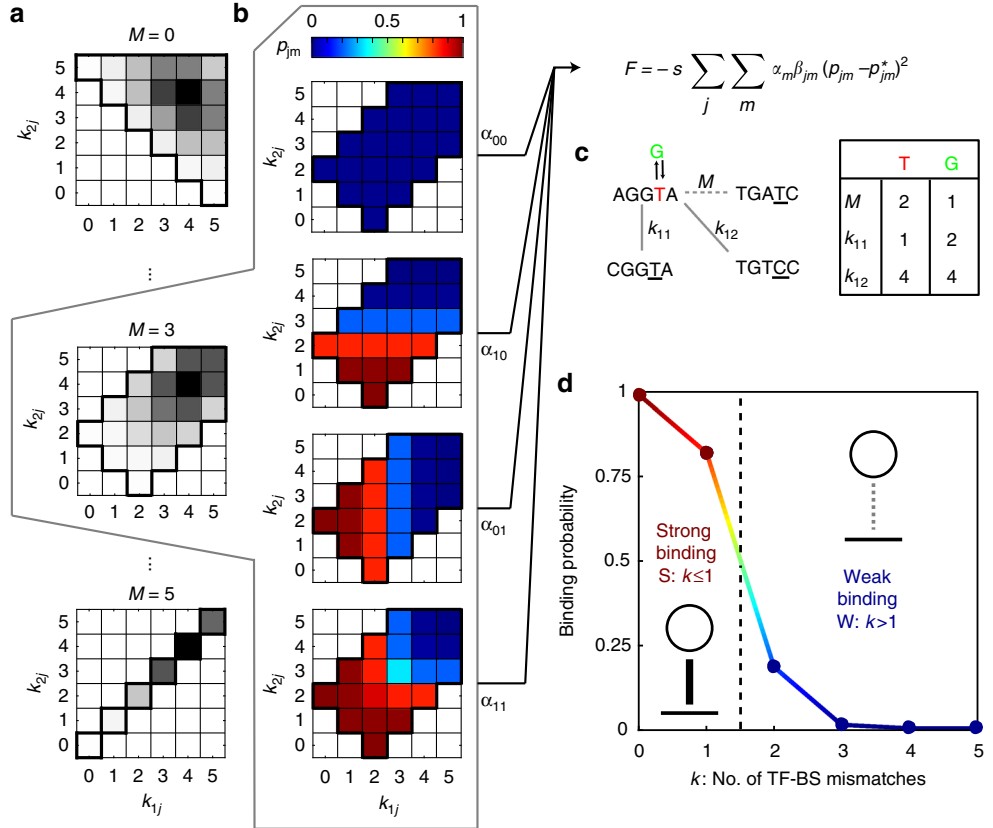

**Fig. 2** Biophysical and evolutionary constraints shape the genotype–phenotype-fitness map after TF duplication. **a** Match, $M$, between transcription factor consensus sequences (here, of length $L = 5$), constrains the possible mismatch values, $k_{1j}$, $k_{2j}$, between the gene's binding site and either TF. For example, when the two TFs are identical ($M = L = 5$, bottom left), they must have equal mismatches with all genes ($k_{1j} = k_{2j}$). Some combinations of mismatches are impossible given $M$ (*white*), while others are realized by different numbers of genotypes (*grayscale*). **b** Expression level (color) for a regulated gene given all mismatch combinations, $k_{1j}$, $k_{2j}$, at $M = 3$. Impossible mismatch combinations are colored white. Each of the four panels shows expression levels in four possible environments, $m = 00, 10, 01, 11$. Fitness $F$ depends on the structure of mismatches **a**, the biophysics of binding **b**, and the frequencies of different environments, $\alpha_m$. Here we choose $\alpha$ so that the marginal probability of each input signal $f_{1,2}$ is always $f_1 = f_2 = \frac{1}{2}$ but the correlation can be varied, and assign weight $\beta_{jm} = 1$ whenever the gene should be induced but is not, and $\beta_{jm} = \frac{1}{2}$ when it is induced when it should not. The general case when $f_1 \neq f_2 \neq 0.5$ is analyzed in Supplementary Note 2. **c** A single point mutation, e.g., a change in one TF's binding specificity from 'T' to 'G', can simultaneously affect the match, $M$, and either increase, decrease, or leave intact the mismatches, $k_{11}$ and $k_{12}$, that determine fitness. **d** TF–BS interactions with mismatch $k$ that is low enough to ensure a high binding probability ($p > 1/2$) are assigned to a "strong binding" phenotype (*solid link*); conversely, $p < 1/2$ is a 'weak binding' phenotype (*dotted link*)

easy to generalize our model to regulation by repressor TFs, a case we explore in Supplementary Note 2.

We consider mutation rates to be low enough that a beneficial mutation fixes before another beneficial mutation arises[51], allowing us to assume that the population is almost always fixed. The probability that the population occupies a particular genotypic state, $P(\mathcal{D}, t)$, evolves according to a continuous-time discrete-space Markov chain that specifies the rate of transition between any two genotypes. The transition rates are a product between the mutation rates between different states and the fixation probability that depends on the fitness advantage a mutant has over the ancestral genotypes[43, 52]. The size of genotype space is high-dimensional but still tractable, because our model only requires us to keep track of mismatches and not full sequences, i.e., to write out the dynamical equations for the reduced-genotypes, $\mathcal{G} = \{M, k_{ij}, \sigma_i\}$. Standard Markov chain techniques can then be used to compute the evolutionary steady state, first hitting times to reach specific evolutionary outcomes, or to perform stochastic simulations (Supplementary Methods).

Figure 2 shows the interplay of biophysical constraints that give rise to a realistic fitness landscape for our problem. Given a match, $M$, between two TF consensus sequences, only certain combinations of mismatches, $(k_{1j}, k_{2j})$, of the TFs with each of the two binding sites are possible. A particular allowed combination can be realized by different numbers of genotypes, as shown in Fig. 2a, providing a detailed account of the entropy of the neutral distribution. For each of the four environments, Eq. (1) predicts gene expression at every pair of mismatch values (Fig. 2b); together with the probabilities of different environments occurring, the gene expression pattern determines the genotypes' fitness, $F$. TF specialization then unfolds on this landscape by different types of mutations (e.g., Fig. 2c). Although the landscape is complex and high-dimensional, it is highly structured and ultimately fully specified by only a handful of biophysical parameters. Furthermore, because of the sigmoidal shape of binding probability as a function of mismatch $k$ (Eq. (1)), it is possible to assign phenotypes of 'strong' and 'weak' binding to every TF–BS interaction, allowing us to depict network interactions graphically, as shown in Fig. 2d, and to classify the possible macroscopic evolutionary outcomes, as we will show next.

**Evolutionary outcomes in steady state.** Evolutionary outcomes in steady state are determined by a balance between selection

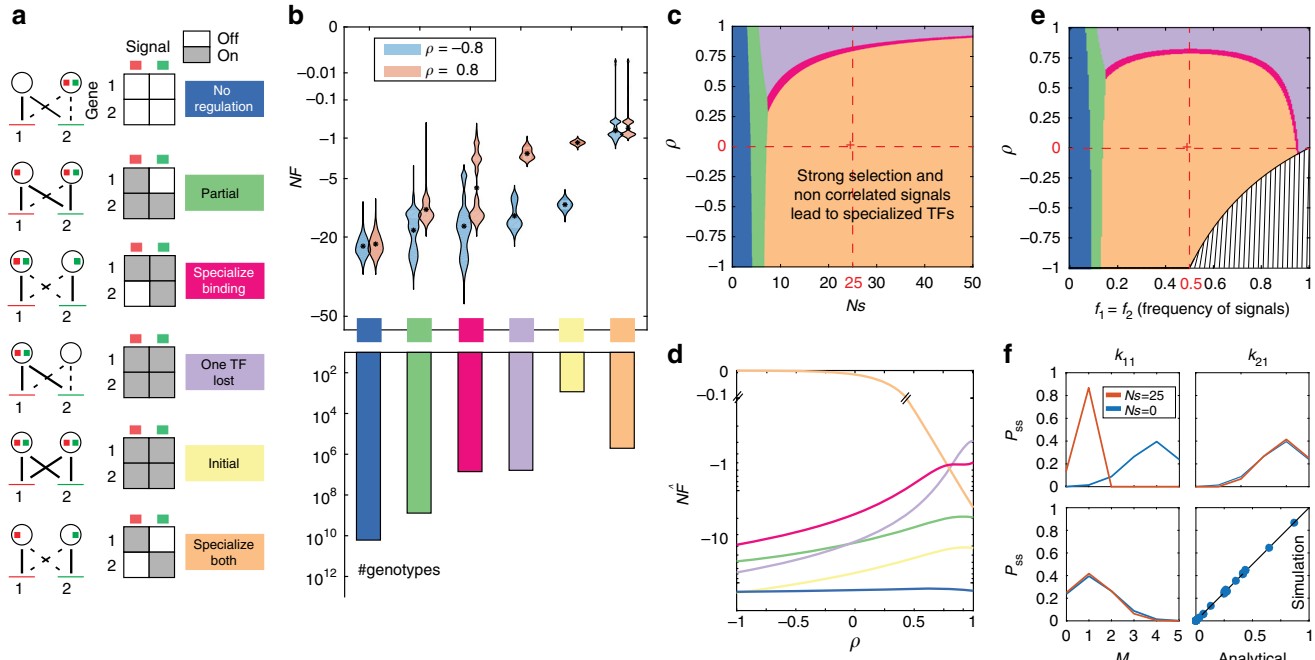

**Fig. 3** Steady state evolutionary outcomes of TF duplication. **a** *Left*: evolutionary macrostates (see text) depicted graphically as network phenotypes with *solid* (*dashed*) lines indicating strong (weak) TF–BS interactions. *Red* and *green squares* in the TFs represent the corresponding signal sensing domains. *Right*: input–output table, where columns represent the presence of either (*red* or *green*) external signal and rows represent the resulting gene activation for each phenotype. **b** (*Top*) Distribution of fitness values across genotypes in each macrostate (color-coded as in **a**), shown as *violin plots*, for two values of signal correlation, $\rho$. *Black dots* = median fitness in the macrostate. (*Bottom*) The number of genotypes in each macrostate (logarithmic scale). **c** Most probable outcome of gene duplication in steady state (color-coded as in **a**), as a function of selection strength, $Ns$, and the correlation between two external signals, $\rho$. **d** Free fitness $\hat{F}$ (at $Ns = 25$) for different macrostates as a function of correlation between signals, $\rho$: for most macrostates, free fitness increases with signal correlation, except for 'No regulation', which is naturally unaffected by it, and 'Specialize Both', which dominates for low correlation values. **e** The dominant macrostate (as in **c**), as a function of the signal frequencies, $f_1$, $f_2$, and the signal correlation, $\rho$, at fixed $Ns = 25$. For simplicity we plot only cases where $f_1 = f_2$. Signals in the *hashed region* are mathematically impossible. **f** Steady state distributions for mismatches ($P_{SS}(k_{ij} \mid \sigma_1 = 10, \sigma_2 = 01)$, upper row) and the match between the two TF consensus sequences ($P_{SS}(M \mid \sigma_1 = 10, \sigma_2 = 01)$, lower left), under strong selection (*red*; at baseline parameters denoted by the *red cross* in **c**) and neutrality (*blue*; Bernoulli distributions). Comparison between analytical calculation and 400 replicates of the stochastic simulation (*lower right*). Here and in subsequent figures, baseline parameter values are $L = 5$, $\epsilon = 3$, $r_S = r_{TF} = 1$

and drift. The steady state distribution over reduced-genotypes is [53]

$$P_{SS}(\mathcal{G}) = P(\mathcal{G}, t \to \infty) = P_0(\mathcal{G}) \exp(2NF(\mathcal{G})), \qquad (3)$$

where $P_0$ is the neutral distribution of genotypes and $N$ is the population size. Eq. (3) is similar to the energy/entropy balance of statistical physics[42, 54], with fitness $F$ playing the role of negative of energy and $\log P_0$ the role of entropy; in our model, both of these quantities are explicitly computable, as is the resulting steady state distribution.

Understanding the high dimensional distribution over genotypes is difficult, but classification of individual TF–BS interactions into "strong" and "weak" ones, as described above, allows us to systematically and uniquely assign every genotype to one of a few possible macroscopic outcomes, or "macrostates," graphically depicted in Fig. 3a and defined precisely in Supplementary Note 1. Thus, in the 'No Regulation' state, input signals are not transduced to the target genes, either because TF–BS mismatches are high and there is no binding or because TFs themselves lose responsiveness to the input signals; in the 'One TF Lost' state, a single TF regulates both genes (as before duplication), while the other TF is lost, i.e., its specificity has diverged so far that it does not bind any of the sites; the 'Specialize Binding' state corresponds to each TF regulating its own gene without cross-regulating the other but the signal sensing domains are not yet signal specific, as they are in the

'Specialize Both', the state which we have defined to have the highest fitness. Finally, the 'Partial' macrostate predominantly features configurations where each of the TFs binds at least one binding site, but one of the TFs still binds both sites or retains responsiveness for both input signals; functionally, these configurations lead to large "crosstalk," where input signals are non-selectively transmitted to both target genes.

Ultimately, these macrostates are the functional network phenotypes that we care about. The number of genotypes in each macrostate, however, can vary by orders of magnitude; for example, the 'No Regulation' state is larger by ~$10^4$ relative to the high-fitness 'Specialize Both' state, for our baseline choice of parameters ($L = 5$, $\epsilon = 3$). Selection can act against this strong entropic bias, and the distribution of fitness values across genotypes within each macrostate is shown in Fig. 3b. Clearly, the mean or median fitness within each macrostate is a poor substitute for the detailed structure of fitness levels that depend nonlinearly on TF–BS mismatches and the degeneracy of the sequence space. Unlike the entropic term in Fig. 3b, fitness also depends on the statistics of the environment, $\alpha_m$, and in particular, the correlation $\rho$ between the two signals. For example, when the signals are strongly correlated, the 'Initial' state right after duplication or the 'One TF Lost' state can achieve quite high fitnesses, since responding to the wrong signal or having a high degree of crosstalk will still ensure largely appropriate gene expression pattern in all likely environments. In contrast, at strong negative correlation, many genotypes in 'Specialize

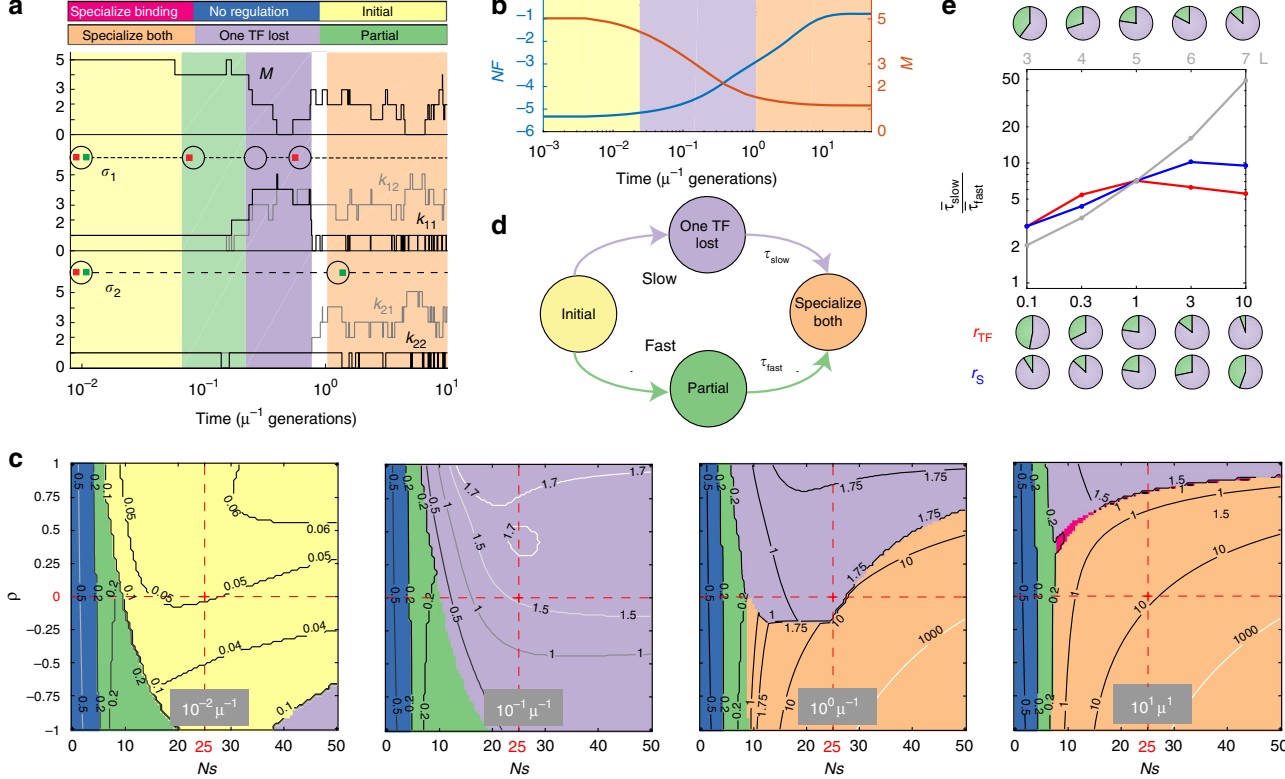

**Fig. 4** Slow and fast pathways to TF specialization. **a** Temporal traces of TF–TF match $M$ (*top*), and TF–BS mismatches $k_{ij}$ (*middle*: TF1, *bottom*: TF2) with the corresponding signal specificity mutations denoted on *dashed lines*, for one example evolutionary trajectory at baseline parameters. Macrostates are color-coded as in the top legend and Fig. 3. **b** Average dynamics of fitness $NF$ (*blue, left scale*) and TF–TF match $M$ (*red, right scale*). For every timepoint, the dominant macrostate is denoted in color. **c** Snapshots of dominant macrostates (at increasing time post-duplication as indicated in the panels), shown for different combinations of selection strength $Ns$ and signal correlation $\rho$ as in Fig. 3. Contours mark dwell times in the dominant macrostates (in units of $\mu^{-1}$). *Red cross*=baseline parameters. **d** Schematic of the two alternative pathways to specialization. $\tau_{\text{slow}}$ and $\tau_{\text{fast}}$ are the total times to specialization for the 'slow' and the 'fast' pathway, respectively. **e** Relative duration of the two pathways, as a function of binding site length $L$ (*gray line, top axis*), TF consensus sequence mutation rate $r_{\text{TF}}$ (*red*), and signal domain mutation rate $r_{\text{S}}$ (*blue, bottom axis*). Pie charts indicate the fraction of slow (*pink*) and fast (*green*) pathways at each parameter value

Binding' and 'Initial' states will suffer a large fitness penalty because their sensing domains are not specialized for the correct signals, while the 'Specialize Both' state will have high fitness regardless of the environmental signal correlation.

How do fitness and entropy combine to determine macroscopic evolutionary outcomes? Fig. 3c shows the most probable macrostate as a function of selection strength and signal correlation (Supplementary Note 2). At weak selection, specific TF–BS interactions cannot be maintained against mutational entropy and the system settles into the most numerous, 'No Regulation' state. Higher selection strengths can maintain a limited number of TF–BS interactions in 'Partial' states. Beyond a threshold value for $Ns$, the evolutionary outcome depends on the signal correlation: when signals are anti-correlated or weakly correlated, the TFs reach the fully specialized state, whereas high positive correlation favors losing one TF and having the remaining TF regulate both genes and respond to both signals. As signal correlation increases, so does the selection strength required to support full specialization. Detailed insight at a fixed value of $Ns$ is provided by plotting the free fitness $\hat{F}$, as in Fig. 3d, which combines the fitness and the entropy of the neutral distribution from Fig. 3b into a single quantity that determines the likelihood of each macrostate given $\rho$; the macrostate with highest free fitness is shown as the most probable outcome in Fig. 3c for $Ns = 25$, but free fitness also allows us to see, quantitatively, how much more likely the dominant macrostate is relative to other outcomes. Figure 3e examines the case where not

only the correlation, $\rho$, but also the frequencies, $f_1$, $f_2$, of encountering both signals are varied: for low frequencies, even selection strength of $Ns = 25$ is insufficient to maintain TF specificity against drift, while for high frequencies and positive correlation one TF is lost while the remaining TF regulates both genes (Supplementary Note 2).

The map of evolutionary outcomes is very robust to parameter variations. The energy scale of TF–DNA interactions is that of hydrogen bonds: $\epsilon \sim 3$ (in $k_B T$ units), consistent with direct measurements. The scale of $C_0$ is set to ensure that consensus sites are occupied at saturation while fully mismatching sites are essentially empty. The only remaining important biophysical parameter is $L$, the length of the binding sites. As expected, increasing $L$ expands the regions of 'No Regulation' and 'Partial' at low $Ns$, due to entropic effects. Surprisingly, however, one can demonstrate that the important boundary between the 'Specialize' and 'One TF Lost' states is independent of $L$; furthermore, the map in Fig. 3c is exactly robust to the overall rescaling of the mutation rate, $\mu$, and even to separate rescaling of individual rates $r_{\text{S}}$, $r_{\text{TF}}$.

TFs can also act as repressors, whereby a gene is active unless a repressor binds its binding site and inhibits its expression. The analysis in that case is very similar to the activator case. The evolutionary outcomes differ only if the penalties $\beta_{jm}$ are non-uniform. Specifically, we consider that unnecessary gene activation incurs a lower penalty $\beta_{jm}$ than does failure to activate a gene when needed. Due to the scarcity of genotypes allowing for

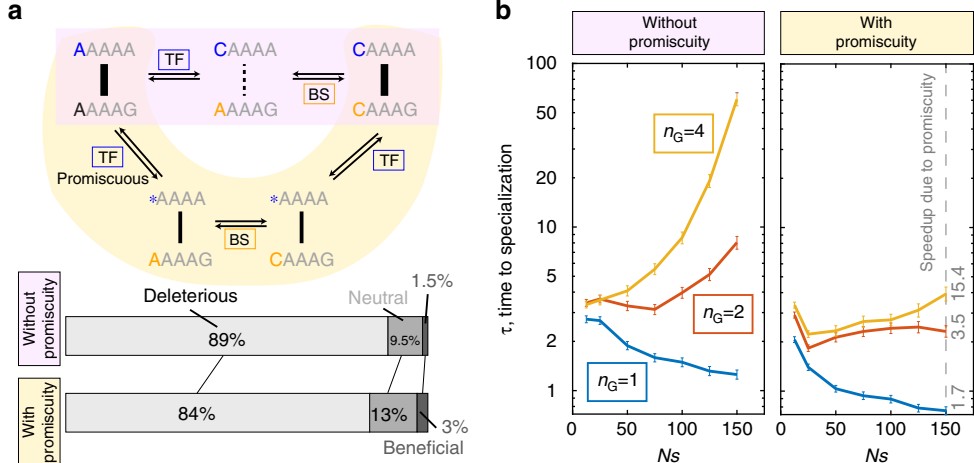

**Fig. 5** Promiscuity-promoting mutations speed up specialization with multiple regulated genes per TF. **a** In the absence of promiscuity-promoting mutations, a compensatory series of point mutations in the TF's consensus (*upper sequence*) and its binding site (*lower sequence*) is needed to maintain TF–BS specificity (top; *light red*). Alternatively, in the presence of promiscuity-promoting mutations in the TF consensus, a position in the TF's recognition sequence (marked by a *star*) can lose and later regain sequence specificity (middle; *light yellow*). Promiscuity decreases the fraction of deleterious mutations along typical pathways to specialization (*bottom*, computed using baseline parameters). **b** Time to specialization as a function of selection strength, $Ns$, without (*left*) and with (*right*) promiscuity promoting mutations in the TF, for different numbers of regulated genes per TF, $n_G$ (*color*). Numbers in *gray* (*right*) denote the speed-up ratio

TF–BS binding compared to the abundance of genotypes for which no binding occurs, this effectively scales the selection pressure, such that higher selection pressure values $Ns$ are required to obtain the more specialized macrostates 'Partial' followed by 'Specialize Both' (Supplementary Note 2).

We compare the steady-state marginal distributions of TF–BS mismatches and the match, $M$, between the two TFs, under strong selection to specialize ($Ns = 25$) vs neutral evolution ($Ns = 0$). Mismatch distributions for $k_{11}$ and $k_{21}$ in Fig. 3f display a clear difference in the two regimes: strong selection favors a small mismatch of the BS with the cognate TF, sufficient to ensure strong binding but nonzero due to entropy, and a large mismatch with the noncognate TF, to reduce crosstalk. Surprisingly, however, the distribution of matches $M$ between two TF consensus sequences shows only a tiny signature of selection, with both distributions peaking around one match. As a consequence, inferring selection to specialize from measured binding preferences of real TFs might not be feasible with realistic amounts of data.

**Evolutionary dynamics and fast pathways to specialization**. Next, we focus on evolutionary trajectories and the timescales to reach the fully specialized state after gene duplication. An example trajectory is shown in Fig. 4a: the two TFs start off identical (with maximal match, $M = L = 5$) until, as a result of the loss of specificity for both signals, TF1 starts to drift, diverging from TF2 (sharply decreasing $M$ in 'One TF Lost' state) and losing interactions with both binding sites. Subsequently TF1 reacquires preference to the red signal, which drives the reestablishment of TF1 specificity for one binding site during a short 'Specialize Binding' epoch, followed quickly by the specialization of TF2 for the green signal at the start of 'Specialize Both' epoch of maximal fitness.

Dynamics of the TF–TF match, $M$, and the scaled fitness, $NF$, become smooth and gradual when discrete transitions and the consequent large jumps in fitness are averaged over individual realizations, as in Fig. 4b. Importantly, we learn that the sequence of dominant macrostates leading towards the final (and steady) state, 'Specialize Both', involves a long intermediate epoch when

the system is in the 'One TF Lost' state. We examine this sequence of most likely macrostates in detail in Fig. 4c, and visualize it analogously to the map of evolutionary outcomes in steady state shown in Fig. 3c. High $Ns$ and correlation ($\rho$) values favor trajectories passing through the 'One TF Lost' state, while intermediate $Ns$ ($5 \lesssim Ns \lesssim 20$) and low correlation values enable transitions through 'Partial' macrostate; along the latter trajectory, the binding of neither TF is completely abolished. Typical dwell times in dominant states, indicated as contours in Fig. 4c, suggest that specialization via the 'One TF Lost' state should be slower than through the 'Partial' state, which is best seen at $t = 1/\mu$, where specialization has already occurred at intermediate $Ns$ and low, but not high, $\rho$ values.

It is easy to understand why pathways towards specialization via the 'One TF Lost' state are slow. As the example in Fig. 4a illustrates, so long as one TF maintains binding to both sites and thus network function (especially when signals are strongly correlated), the other TF's specificity will be unconstrained to neutrally drift and lose binding to both sites, an outcome which is entropically highly favored. After the TF's sensory domain specializes, however, the binding has to re-evolve essentially from scratch in a process that is known to be slow[45] unless selection strength is very high. In contrast to this 'Slow' pathway, the 'Fast' pathway via the 'Partial' state relies on sequential loss of "crosstalk" TF–BS interactions, with the divergence of TF consensus sequences followed in lock-step by mutations in cognate binding sites. Specifically, the likely intermediary of the fast pathway is a 'Partial' configuration in which the first TF responds to both signals but only regulates one gene, whereas the second TF is already specialized for one signal, but still regulates both genes.

The fast and the slow pathways are summarized in Fig. 4d. A detailed analysis (Supplementary Note 4) reveals how different biophysical and evolutionary parameters change the relative probability and the average duration (Fig. 4e) of both pathways. For example, increasing the length, $L$, of the binding sites favors the slow pathway as well as drastically increases its duration, leading to very slow evolutionary dynamics. In contrast, time to specialize via the fast pathway is unaffected by an increase in $L$. Increasing the rate of TF-specificity-affecting mutations, $r_{TF}$, has

a qualitatively similar effect, while increasing the mutation rate affecting the sensory domain, $r_S$, favors the fast pathway. Indeed, in the limit when $r_S$ is much larger than the other two mutation rates, the sensing domain specializes almost instantaneously, making the complete loss of binding by either TF very deleterious and thus avoiding the 'One TF Lost' state; the adaptation dynamics is initially rapid, with binding sites responding to diverging TF consensus sequences, and subsequently slow, when TF consensus sequences further minimize their match, $M$, in a nearly neutral process.

**Promiscuity-promoting mutations**. Typically, each TF must regulate more than one target gene. As the number of regulated genes per TF ($n_G/n_{TF}$) increases, intuition suggests that the evolution of the TF's consensus sequence should become more and more constrained: while a mutation in an individual binding site can lower the total fitness by increasing mismatch and thereby impeding TF–BS binding, a single mutation in the TF's consensus has the ability to simultaneously weaken the interaction with many binding sites, leading to a high fitness penalty. Our analysis of the biophysical fitness landscape confirmed that the landscape gets progressively more frustrated as the number of regulated genes per TF increases, due to the explosion of constraints that TFs have to satisfy to ensure the maintenance of functional regulation (Supplementary Note 5). Consequently, one can expect extremely long times to specialization. How can it nevertheless proceed at observable rates?

Energy matrices for many real TFs display 'promiscuous' specificity where, at a particular position within the binding site, binding to multiple nucleotides is equally preferable. We wondered how our findings would be affected if consensus sequence specificity of the TFs could pass through such intermediate promiscuous states. Figure 5a shows how TF consensus sequence and the corresponding binding site can co-evolve using point mutations, or using the new "promiscuity-promoting" mutation type for the TF: promiscuity-promoting mutation renders one position in the recognition sequence of the TF insensitive to the corresponding DNA base in the binding site (Supplementary Note 6). Evolutionary pressure on the binding sites is therefore temporarily relieved, until the specificity of the TF is re-established by a back mutation. Without promiscuity-promoting mutations, TF–BS co-evolution must proceed in a tight sequence of compensatory mutations; with promiscuity-promoting mutations, such a precise sequence is no longer required, although one extra mutation is needed to reestablish high TF–BS specificity. With promiscuity, the fraction of deleterious mutations along the evolutionary path towards specialization is reduced, an effect that grows stronger with increasing $L$. As shown in Fig. 5b, this has drastic effects on the time to specialization. Without promiscuity, increasing the selection strength, $Ns$, decreases the required time when each TF regulates one gene, as expected for a landscape with large neutral plateaus but with no fitness barriers. For $n_G > 2$, however, the landscape develops barriers that need to be crossed, and evolutionary time starts increasing with $Ns$. In contrast, promiscuity enables fast emergence of TF specialization even with multiple regulated genes in a broad range of evolutionary parameters (although there are also costs due to high promiscuity).

## Discussion

The role that the shape of a fitness landscape plays for the dynamics and the final outcomes of evolution has been appreciated in population genetics for a long time. This has stimulated a large body of theoretical research into evolution on

toy model landscapes[55, 56], as well as motivated efforts to map out real, small-scale landscapes experimentally. For limited classes of problems, mostly those involving molecular recognition, biophysical constraints are informative enough to permit computational exploration of complex landscapes. Such is the case for the secondary structure of RNA[57], antibody–antigen interactions[58], protein–protein interactions[59], and TF–DNA binding[60], explored here. We exploit this prior knowledge to construct a fitness landscape for a more complicated evolutionary event, the specialization of two TFs after duplication, a key evolutionary step by which gene regulatory networks expand. The biophysical model naturally captures a number of essential features, without having to introduce them 'by hand': the fact that specialization is driven by avoidance of regulatory crosstalk; the importance of the mutational entropy; the dependence on number of downstream genes; the existence of transient network configurations preceding specialization, which crucially impact dynamics; and the importance for evolutionary outcomes of the statistical properties of the signals that TFs respond to. Importantly, the expressive power of our framework does not come at increased modeling cost: while complex, the fitness landscape is still determined only by a few, mostly known, parameters, and an exponentially large space of genotypes can be systematically coarse grained to a small set of functional network phenotypes. This combination of biophysical and co-evolutionary approaches is applicable generally to the evolution of molecular interactions, e.g., in protein interaction networks.

In steady state, our results robustly identify correlation between the environmental signals that drive TFs as a key determinant for specialization, as shown in Fig. 3c–e. Unless the new signal, for which a post-duplication TF can specialize, is sufficiently independent (uncorrelated) from the existing signals that the regulatory network processes, one TF copy will be lost due to drift. As a consequence, the effective dimensionality of environmental signals dictates the complexity of genetic regulatory networks[61], reminiscent of information-theoretic tradeoffs in sensory neuroscience[62]; in evolutionary terms, selection to maintain complex regulation needs to withstand the mutational flux into vastly more numerous but less functional network phenotypes. Recently, it has been shown that finite biochemical specificity also limits the complexity of genetic regulatory networks[63]; an interesting direction for future research is to understand how the balance between regulatory crosstalk, environmental signal statistics, and evolutionary constraints ultimately determines the number of TFs that can be stably maintained. A related question concerns the expected match between pairs of TFs in a large network as a signature of selection for specialized function; for an isolated pair of TFs, our results in Fig. 3f predict only a tiny deviation from neutrality.

Timescales and pathways to specialization are completely shaped by the properties of the biophysical fitness landscape, and thus cannot be captured by simple allelic models that ignore the topology of the sequence space (Supplementary Note 7). We show that the fast pathway to specialization transitions through 'Partial' states where neither of the two TFs completely loses binding. Interestingly, it is exactly the existence of crosstalk interactions that permits fast adaptation via these transient states, by maintaining the network function through one TF, while the other is free to diverge in a series of mutations to the TF and its future binding site[64]. Crosstalk thus enables some amount of network plasticity during early adaptation, yet is ultimately selected against, when TFs become fully specialized[65, 66]. In the protein–protein-interaction literature, 'Partial' states are sometimes referred to as promiscuous states, and they have been suggested as evolutionarily accessible intermediaries that relieve

the two interacting molecules of the need to evolve in a tight (and likely very slow) series of compensatory mutations[67]. In contrast to the fast pathway, the slow pathway involves a complete loss of TF–BS binding interactions; the long timescale emerges from long dwell times while the TF and the binding sites evolve in a nearly neutral landscape before TF–BS specificity is reacquired. Long-binding sites and (perhaps counter-intuitively) fast TF mutation rates favor the slow pathway, while fast sensing domain mutation rates favor the fast pathway.

The situation changes qualitatively when each TF regulates more genes[68]. On the one hand, entropy makes pathways that pass through the 'One TF Lost' state dynamically uncompetitive, as multiple binding sites would have to emerge de novo to reestablish interactions with a diverged TF. This would favor fast pathways through 'Partial' states. On the other hand, the biophysical fitness landscape develops frustration (or sign epistasis) as $n_G > 2$ and the timescales to specialization lengthen with increasing selection strength when passing through 'Partial' states. We demonstrate that frustration is relieved by promiscuity-promoting mutations in the TF, enabling fast emergence of specialization even with multiple-regulated genes.

Recent experimental works have demonstrated how a combination of cis and trans mutations can rewire gene regulatory networks allowing for the emergence of new functions via transient and promiscuous configurations, in accordance with our model[15]. While we focused on a specific evolutionary scenario involving TF duplication, gene regulatory networks can rewire in numerous other ways. For example, Sayou et al. studied the evolution of TF–DNA binding specificity while the TF remains present in a single copy[14]. Duplicated TFs can also be re-used in ways that are different from what we considered[26]. Our results do, however, make predictions for expected timescales to reach different network configurations after gene duplication, which can be compared to bioinformatic data; alternatively, genomic data on TF duplication events could be used to infer selection pressures favoring regulatory divergence.

Taken together, our results paint a picture of TF specialization that most likely proceeds through intermediate states with high crosstalk, in which one TF has already specialized for its input signals but not yet for the target genes, while the other TF is not yet specialized for the input signals but only regulates one gene. In addition, these intermediate states are likely to be more promiscuous, binding different sites with the same affinity, with the promiscuity reverting to specific binding towards the end of specialization. This picture is qualitatively different from the paradigmatic idea of a simple and sequential progression of compensatory mutations in the TF and its binding sites[46, 69]. It depends fundamentally on the biophysical model of TF–BS interactions, predicts significantly faster specialization times, as well as the existence of promiscuous TF variants that are starting to be observed in genomic analyses of duplication-specialization events[14, 15].

## Methods

We consider mutation rates to be low enough that a beneficial mutation fixes before another beneficial mutation arises[51], allowing us to assume that the population is almost always captured by a single genotype. The probability that the population occupies a particular genotypic state, $P(\mathcal{D}, t)$, evolves according to a continuous-time discrete-space Markov chain. Transition rates between states are a product between the mutation rates between different genotypes and the fixation probabilities that depend on the fitness advantage a mutant has over the ancestral genotype[43, 52], $r_{xy} = 2N\mu_{xy}\Phi_{y\to x}$, where $N$ is the population size, $\mu_{xy}$ is the mutation rate from genotype $y$ to $x$, and $\Phi_{y\to x}$ is the probability of fixation of a single copy of $x$ in a population of $y$. Our model only requires us to keep track of mismatches and not full sequences (i.e., the reduced-genotypes, $\mathcal{G} = \{M, k_{ij}, \sigma_i\}$), which significantly reduces the genotype space dimensionality. This framework allows for calculation of the steady state distribution of genotypes, or reduced-genotypes Eq. (3) and classification of genotypes into relevant macrostates.

To calculate the neutral distribution $P_0$ of the reduced-genotypes (distribution in the absence of selection), we enumerate the number of possible BS sequences $j$ that have mismatch values $(k_{1j}, k_{2j})$ with respect to two TFs that match each other at $M$ out of $L$ consensus positions:

$$N_{\text{seq}}(k_1, k_2 | M) = \sum_{j_0 = j_0^{\min}}^{j_0^{\max}} \binom{M}{j_0} 3^{M - j_0} \binom{L - M}{L - j_0 - k_1} \binom{j_0 + k_1 - M}{L - j_0 - k_2} 2^{k_1 + k_2 + 2j_0 - L - M}$$

$$j_0^{\min} = \max\left(\max(0, M - \min(k_1, k_2)), \left\lceil \frac{L + M - k_1 - k_2}{2} \right\rceil\right)$$

$$j_0^{\max} = \min(M, L - \max(k_1, k_2)) \tag{4}$$

The neutral distribution (up to proportionality constant) equals

$$P_0(x) \sim N_{\text{seq}}(k_{11}, k_{21} | M) N_{\text{seq}}(k_{12}, k_{22} | M) \binom{L}{M} 3^{L - M}. \tag{5}$$

We iterate this calculation for various parameter combinations ($Ns$, $\rho$, $f_{1,2}$). For each, we determine the most probable macrostate at steady state (Supplementary Note 2) as illustrated in Fig. 3.

To determine evolutionary dynamics we numerically integrate $P(\mathcal{G}, t)$ in time-steps corresponding to one generation $t_g$:

$$P(\mathcal{G}, t + t_g) = P(\mathcal{G}, t) + \mathbf{R} t_g P(\mathcal{G}, t), \tag{6}$$

where $R$ is the Markov chain transition matrix. Again, at every time-point we determine the most probable macrostate (Supplementary Note 2), as illustrated in Fig. 4.

To follow different pathways to specialization and the timescale to reach each, we calculate mean first hitting time $T_{S \leftarrow x}$ from any reduced-genotype $x$, to a subset of reduced-genotypes $S$, by using the recursive equation

$$T_{S \leftarrow x} = t_g + \sum_y a_{yx} T_{S \leftarrow y}, \tag{7}$$

where $a_{yx}$ are the elements of the transition probability matrix $\mathbf{A} = \mathbf{I} + \mathbf{R} t_g$. In particular, we consider subsets $S_z$ of genotypes that belong to a particular macrostate $z$, and compute the mean first hitting times, $T_{S_z \leftarrow x}$, to this macrostate. Time to specialization, $\tau$, is the time to reach 'Specialize Both' macrostate. We also calculate "dwell times", $t^{\text{dwell}}(z)$ by using a similar procedure. Dwell time in a particular macrostate $z$, is the mean (taken over all the genotypes in $z$, $S_z$) first hitting time to any other macrostate, starting from $S_z$.

We supplement these analytical solutions by stochastic simulations. Using Gillespie algorithm[70], we draw random times in which substitutions between distinct (reduced-)genotypes occurred. At each simulation run a we generate a specific evolutionary trajectory. By repeating this procedure numerous times, we obtain statistics over the distributions and evolutionary pathways. We use stochastic simulations to either validate the analytical calculations or substitute them when they are hard. That is the case, for example, for calculation of mean hitting time to a particular macrostate conditioned on not hitting another macrostate before, as in $\{\tau_{\text{fast}}\}$ and $\{\tau_{\text{slow}}\}$ (Fig. 4). More details about the methods are given in Supplementary Methods.

**Data availability**. The authors declare that all data supporting the findings of this study are available within the article and its Supplementary Information file.

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

## Acknowledgements

We thank the People Programme (Marie Curie Actions) of the European Union's Seventh Framework Programme (FP7/2007–2013) under REA grant agreement Nr. 291734 (T.F.), ERC grant Nr. 250152 (N.B.), and Austrian Science Fund grant FWF P28844 (G.T.).

## Author contributions

T.F., R.P., N.H.B., and G.T. designed the study. T.F. and R.P. carried out the calculations and analysis. T.F. and G.T. wrote the paper.

## Additional information

**Competing interests:** The authors declare no competing financial interests.

