## [Peer Review file · Nature Communications]

Reviewers' comments:

Reviewer #1 (Remarks to the Author):

Friedlander & Prizak et al. study the coevolution of transcription factors (TFs) and their binding sites upon TF duplication. They investigate how such coevolution is affected by several parameters, including the number of genes each TF regulates, the length and specificity of the binding sites, and how the input signals that activate the TFs are correlated. To do so, they consider a model that synthesizes the fundamentals of traditional gene duplication-differentiation models with a classic biophysical model of TF-DNA interactions. This synthesis results in what the authors call a "biophysically realistic fitness landscape," which improves upon previous modeling work in several ways, including the incorporation of a many-to-one mapping from genotypes to functional architectures and the possibility to study the important influence of crosstalk. Analysis and simulation of this model uncover evolutionary trajectories toward TF specialization that cannot be found using either of the two modeling components alone. For example, the evolutionary transition through the ``partial' macrostate is not observed in a simple biallelic-like model.

The main results of this study are that (1) there is an entropic bias toward certain macrostates that selection must overcome for specialization to evolve, (2) specialization can evolve through multiple evolutionary trajectories that differ in their timescales, (3) specialization is more likely to evolve when the input signals are uncorrelated or anti-correlated, and that (4) promiscuity-promoting mutations --- those that decrease specificity at a single locus of a binding site --- can increase the rate of evolution toward TF specialization, overcoming the constraints imposed by TFs' having to regulate many genes.

This paper is timely, convincing, highly original, and goes far beyond the author's previous work on the topic. And like their previous work, this paper will certainly help shape the future of the field. Moreover, it will be relevant to a broad audience --- not only those interested in gene regulation, but also those interested in gene duplication and in the evolution of regulatory networks. Importantly, the supplementary material provides enough detail to ensure the reproducibility of the work and that the primary results are not sensitive to the baseline parameter combination studied in the main text.

This is one of the most polished papers I've ever reviewed, and as such I have only one very minor comment:

- The evolutionary model assumes that the duplicate TF is fixed in the population. I think it is worth discussing how such fixation may occur, e.g., via dosage compensation (Lan & Pritchard, Science, 2016).

Trivia:

p.16,1303-304: Please provide references for antibody-antigen interactions, protein-protein interactions, and transcription factor-DNA binding. Some suggestions: antibody-antigen interactions (Adams et al., eLife, 2016), protein-protein interactions (Podgornaia & Laub, Science, 2015), transcription factor-DNA binding (Aguilar-Rodriguez et al., Nature Ecology & Evolution, 2017)

p.17,1322: Please provide a reference for "information-theoretic tradeoffs in sensory neuroscience."

Some references are provided out-of-order. E.g., p.135: [22,3], p.3178: [47,48,42,34]

In the bibliography, some of the article titles have every word capitalized, whereas others do not.

p.14,252: (4E -> (4E)

p.22,1470: "MutationSelection" -> "Mutation-Selection" Also in the Supp.

Supp, before eq S5: "Tje" -> "The"

Supp, before "Time to specialization" section: "becomes be" -> "becomes"

Reviewer #2 (Remarks to the Author):

The paper by Friedlander et al treats the evolution of regulatory networks, using a joint approach of biophysical and evolutionary modelling. The model links external regulatory cues with evolution of regulated factors and their binding sites; the fitness function of this model is built on the correlation of the input signals and the expression of their target genes.

The strength and novelty of the approach is in the integration of different physiological units, which are distinct targets of natural selection, into a unified conceptual and modelling framework. The most interesting result is a quantitative insight on possible pathways of regulatory differentiation involving pleiotropic transcription factors (which are under strong selection to maintain existing functions). The paper is overall sound and well written. However, a few points are not yet fully convincing and should be addressed in a revised version:

1. The presentation of the results can be improved in a number of places, with a particular eye on the broad readership of Nature Communications.

(a) The introduction could say more clearly what are the key open biological questions the paper wants to address: what are likely pathways and tempo of regulatory differentiation. It could also walk the reader through the different steps of the paper: a simplified case of just two regulated genes, and an extension to larger numbers of genes.

(b) The figures try to convey, in my opinion, a bit too much detail, and many readers will be a bit lost. An intuitive option that could be tried is a figure that follows the physiological order of events: how an input signal (say, green on/off) propagates through the system under different scenarios and induces the downstream units with probabilities shown by color coding. More importantly, some key data are difficult to read off from the figures. For example, the balance between fitness and entropy is contained in a somewhat convoluted way in Fig. 3A&B. It might be more

informative to plot directly the free fitness as a function of control parameters (e.g. ρ or N), so that changes in the ranking of macrostates become explicit. Minor remark: ref. 41, where the fitness entropy balance was first discussed in the context of TF binding, should be cited along with ref. 53 below eq. 3.

2. An intriguing and potentially far-reaching insight of the paper is the link between the input signal correlation and the complexity of the regulatory output. However, this result is based on some assumptions that are not explicitly stated. Each signal by itself has an important characteristic: the fraction q of time that it is in the on/off state. This fraction influences the likely regulatory mode of the target genes: regulation by activation or by repression. This begs the questions: (a) How do the authors' results depend on q , and (b) do they depend on regulation by activation or are they more robust?

3. Recent experimental work (e.g. ref. 12, 14, 15 and recent follow-up work) is briefly discussed at the end but I think the links should be strengthened: what is the overlap of existing data with the results of this paper, what are new predictions on this kind of experiments.

4. Some more comment on the methods (numerics vs. analytical estimates) are warranted in the main text, for example regarding the statistics of macrostates and the analysis of pathways and times (line 216 and below).

Response to reviewer comments

Reviewer comments are *italicized*. Responses are in normal font.

Reviewer #1

Friedlander & Prizak et al. study the coevolution of transcription factors (TFs) and their binding sites upon TF duplication. They investigate how such coevolution is affected by several parameters, including the number of genes each TF regulates, the length and specificity of the binding sites, and how the input signals that activate the TFs are correlated. To do so, they consider a model that synthesizes the fundamentals of traditional gene duplication-differentiation models with a classic biophysical model of TF-DNA interactions. This synthesis results in what the authors call a "biophysically realistic fitness landscape," which improves upon previous modeling work in several ways, including the incorporation of a many-to-one mapping from genotypes to functional architectures and the possibility to study the important influence of crosstalk. Analysis and simulation of this model uncover evolutionary trajectories toward TF specialization that cannot be found using either of the two modeling components alone. For example, the evolutionary transition through the 'partial' macrostate is not observed in a simple biallelic-like model.

The main results of this study are that (1) there is an entropic bias toward certain macrostates that selection must overcome for specialization to evolve, (2) specialization can evolve through multiple evolutionary trajectories that differ in their timescales, (3) specialization is more likely to evolve when the input signals are uncorrelated or anti-correlated, and that (4) promiscuity-promoting mutations --- those that decrease specificity at a single locus of a binding site --- can increase the rate of evolution toward TF specialization, overcoming the constraints imposed by TFs' having to regulate many genes. This paper is timely, convincing, highly original, and goes far beyond the author's previous work on the topic. And like their previous work, this paper will certainly help shape the future of the field. Moreover, it will be relevant to a broad audience --- not only those interested in gene regulation, but also those interested in gene duplication and in the evolution of regulatory networks. Importantly, the supplementary material provides enough detail to ensure the reproducibility of the work and that the primary results are not sensitive to the baseline parameter combination studied in the main text.

This is one of the most polished papers I've ever reviewed, and as such

We thank the reviewer for these warm and encouraging words.

I have only one very minor comment:

- The evolutionary model assumes that the duplicate TF is fixed in the population. I think it is worth

discussing how such fixation may occur, e.g., via dosage compensation (Lan & Pritchard, Science, 2016).

We thank the reviewer for highlighting this point. We now refer to evolutionary preservation of the duplicates in the introduction (the new text is in bold font):

“The main avenue for such expansions are gene duplications~\cite{ohno_evolution_1970,magadum_gene_2013,yona_chromosomal_2012}. **Rapid weakening of expression of the duplicates~\cite{lan_coregulation_2016} or alternatively selection to increase expression~\cite{conant_dosage_2014,loehlin_expression_2016} facilitate the preservation and fixation of duplicates. Gene duplications generate** extra copies of the TFs and thus provide the “raw material” for evolutionary diversification. Subsequent specialization of TFs often involves divergence in both their inputs (e.g., ligands) and outputs (regulated genes)~\cite{wray_evolutionary_2007,wittkopp_cis-regulatory_2012}.”

Trivia:

p.16,1303-304: Please provide references for antibody-antigen interactions, protein-protein interactions, and transcription factor-DNA binding. Some suggestions: antibody-antigen interactions (Adams et al., eLife, 2016), protein-protein interactions (Podgornaia & Laub, Science, 2015), transcription factor-DNA binding (Aguilar-Rodriguez et al., Nature Ecology & Evolution, 2017)

We thank the reviewer for pointing that out. We added these relevant references.

p.17,1322: Please provide a reference for "information-theoretic tradeoffs in sensory neuroscience."

We have added a reference to Tkacik et al, “Optimal population coding by noisy spiking neurons” PNAS 107: 14419 (2010).

Some references are provided out-of-order. E.g., p.135: [22,3], p.3178: [47,48,42,34]

We corrected that.

In the bibliography, some of the article titles have every word capitalized, whereas others do not.

Corrected.

p.14,252: (4E -> (4E)

Corrected.

p.22,1470: "MutationSelection" -> "Mutation-Selection" Also in the Supp.

Corrected.

Supp, before eq S5: "Tje" -> "The"

Corrected.

Supp, before "Time to specialization" section: "becomes be" ->"becomes"

Corrected.

Reviewer #2 (Remarks to the Author):

The paper by Friedlander et al treats the evolution of regulatory networks, using a joint approach of biophysical and evolutionary modelling. The model links external regulatory cues with evolution of regulated factors and their binding sites; the fitness function of this model is built on the correlation of the input signals and the expression of their target genes.

The strength and novelty of the approach is in the integration of different physiological units, which are distinct targets of natural selection, into a unified conceptual and modelling framework. The most interesting result is a quantitative insight on possible pathways of regulatory differentiation involving pleiotropic transcription factors (which are under strong selection to maintain existing functions). The paper is overall sound and well written. However, a few points are not yet fully convincing and should be addressed in a revised version:

1. The presentation of the results can be improved in a number of places, with a particular eye on the broad readership of Nature Communications.

(a) The introduction could say more clearly what are the key open biological questions the paper wants to address: what are likely pathways and tempo of regulatory differentiation. We thank the reviewer for this comment. We augmented the paragraph in the introduction that details the biological questions, which now reads (new text in bold font):

“Immediately following a duplication event, molecular recognition between TFs, their input signals, and their binding sites is specific but undifferentiated between the two TF copies. Under selection to specialize, recognition sequences and ligand preferences of the two TFs can diverge, but only if some degree of matching between TFs and their binding sites is continually retained to ensure network function. Binding sites are thus forced to coevolve in tandem with the TFs. Yet little is known about the **resulting limits to evolutionary outcomes, the likelihood of potential evolutionary trajectories, and the relevant timescales; specifically, it is unclear how these quantities of interest depend on important parameters**, such as the number of regulated genes, the length and specificity of the binding sites, the correlations between the input signals, etc.”

It could also walk the reader through the different steps of the paper: a simplified case of just two regulated genes, and an extension to larger numbers of genes.

At the end of the introduction we now walk the reader through the paper structure:

“The structure of the paper is as follows. We first introduce the basic model with two TFs and two regulated genes, and analyze its steady state distribution of outcomes, showing that the huge genotypic space maps to very few phenotypes. We next analyze the possible dynamical trajectories and timescales leading to each phenotype. Finally, we extend the basic model to a larger number of

regulated genes and study the effect of "promiscuity-promoting mutations," i.e., mutations that render TFs less specific for their binding sites."

(b) The figures try to convey, in my opinion, a bit too much detail, and many readers will be a bit lost. An intuitive option that could be tried is a figure that follows the physiological order of events: how an input signal (say, green on/off) propagates through the system under different scenarios and induces the downstream units with probabilities shown by color coding.

We thank the reviewer for this comment. We have modified Fig. 1, which now includes an introductory panel (a) that schematizes the physiological order of events in signal transduction: signals that impinge on the cell interact with TFs in the cell to activate them, which in turn leads to TF binding and consequently control of gene expression. This figure now introduces the key concepts of our paper in a cellular context.

More importantly, some key data are difficult to read off from the figures. For example, the balance between fitness and entropy is contained in a somewhat convoluted way in Fig. 3A&B. It might be more informative to plot directly the free fitness as a function of control parameters (e.g. ρ or N), so that changes in the ranking of macrostates become explicit.

We have modified Fig. 3, which now also shows the free fitness as a function of the signal correlation ρ (panel 3d). We would like to retain the separate plots of fitness and entropy that, when combined, give free fitness, to show how the components of free fitness originate from our biophysical model. The new free fitness panel clearly explains the different dependencies of the various macrostates on ρ and shows the ranking of macrostates. We point out that it thus provides a more detailed insight compared to Fig 3c, which only shows the dominant macrostate.

Minor remark: ref. 41, where the fitness entropy balance was first discussed in the context of TF binding, should be cited along with ref. 53 below eq. 3.

Done.

2. An intriguing and potentially far-reaching insight of the paper is the link between the input signal correlation and the complexity of the regulatory output. However, this result is based on some assumptions that are not explicitly stated. Each signal by itself has an important characteristic: the fraction q of time that it is in the on/off state. This fraction influences the likely regulatory mode of the target genes: regulation by activation or by repression. This begs the questions: (a) How do the authors' results depend on q ,

We have added another panel to Fig. 3 (Fig 3e) where we show the most probable macrostate for different frequencies f_1 and f_2 (q) of the two signals, where $f_1=f_2$. Most of the phase space is very similar to the baseline case (with $f_1=f_2=0.5$). The only difference we see is the expansion of the region in which one TF is lost (which now includes lower ρ values for high f_1 and f_2). In Supplementary Note 2 we study the more general case when occurrences of the two signals are unequal $f_1 \neq f_2$.

and (b) do they depend on regulation by activation or are they more robust?

We have added a new section to the Supplementary Note 2 where we study the case of regulation by repressors. In brief, we find the evolution of the repressor case very similar to the activator case studied in the main text. The only difference comes about if we consider a lower fitness cost to activating an unnecessary gene compared to avoidance of activation of a necessary one (β values for Eq 2 in the main text).

This is because we assume different default states of the downstream genes in the activator vs repressor cases: if the TFs are activators the genes are by default inactive (unless activated by a TF), whereas if the TFs are repressors the genes are by default active unless repressed by TFs. Hence, a failure of a TF to bind the regulatory binding sites is penalized differently for activators and repressors. This can be roughly described as a scaling of the selection intensity s (see Supplementary Fig. 13).

We find that for repressor-TFs the dependence of the macrostate plot on signal frequencies is the mirror image of the activator-TF case: If both signal frequencies are very low, in the activator case the TF binding cannot be maintained against drift and ‘No regulation’ state emerges, whereas in the repressor case it means that a single repressor is sufficient (resulting in ‘One TF Lost’ state). If both signal frequencies are high, the situation is inverted: for the activator-TF case the ‘One TF Lost’ state is favored, while for the repressor the binding cannot be maintained and the ‘No regulation’ state emerges (Supplementary Fig. 14, compare to Fig. 3e in the main text). This activator/repressor symmetry is expected, but nice to see confirmed.

3. Recent experimental work (e.g. ref. 12, 14, 15 and recent follow-up work) is briefly discussed at the end but I think the links should be strengthened: what is the overlap of existing data with the results of this paper, what are new predictions on this kind of experiments.

We have added the following text to the discussion:

“Recent experimental works have demonstrated how a combination of \emph{cis} and \emph{trans} mutations can rewire gene regulatory networks allowing for the emergence of new functions via transient and promiscuous configurations, in accordance with our model~\cite{pougach_duplication_2014}. While we focused on a specific evolutionary scenario involving TF duplication, gene regulatory networks can rewire in numerous other ways. For example, Sayer~\etal~studied the evolution of TF-DNA binding specificity while the TF remains present in a single copy ~\cite{sayer_promiscuous_2014}. Duplicated TFs can also be re-used in ways that are different from what we considered~\cite{perez_how_2014}. Our results do, however, make predictions for expected timescales to reach different network configurations after gene duplication, which can be compared to bioinformatic data; alternatively, genomic data on TF duplication events could be used to infer selection pressures favoring regulatory divergence.”

4. Some more comment on the methods (numerics vs. analytical estimates) are warranted in the main text, for example regarding the statistics of macrostates and the analysis of pathways and times (line 216 and below).

We have added a “Methods” section to the main text to briefly describe our techniques (which are explained in more detail in Supplementary Notes).

REVIEWERS' COMMENTS:

Reviewer #2 (Remarks to the Author):

The authors have addressed my recommendations in a very careful way, the presentation has become clearer and is now entirely convincing. One more small suggestion: it would be appropriate if the convincing results on regulation by repression make their way into a summary paragraph in the main text.

Otherwise I recommend publication of the manuscript in its present form.

Response to reviewer comments

Reviewer comments are *italicized*. Responses are in normal font.

Reviewer #2

The authors have addressed my recommendations in a very careful way, the presentation has become clearer and is now entirely convincing. One more small suggestion: it would be appropriate if the convincing results on regulation by repression make their way into a summary paragraph in the main text.

Otherwise I recommend publication of the manuscript in its present form.

We thank the reviewer for the assessment of our revised manuscript and for the encouraging review.

We have added the following text to the main paper to describe the results of our model when TFs are repressors (lines 219-226):

“Transcription factors can also act as repressors, whereby a gene is active unless a repressor binds its binding site and inhibits its expression. The analysis in that case is very similar to the activator case. The evolutionary outcomes differ only if the penalties β_{jm} are non-uniform. Specifically, we consider that unnecessary gene activation incurs a lower penalty β_{jm} than does failure to activate a gene when needed. Due to the scarcity of genotypes allowing for TF-BS binding compared to the abundance of genotypes for which no binding occurs, this effectively scales the selection pressure, such that higher selection pressure values N_s are required to obtain the more specialized macrostates 'Partial' followed by 'Specialize Both' (see Supplementary Note 2).”

In the description of the model we also mention the possibility that TFs are repressors (lines 123-127):

“Importantly, selection does not directly depend on the TFs, but only on the expression state of the genes they regulate; genes, however, can only be expressed when TFs bind to proper binding sites, implicitly selecting on TFs. For this reason it is also very easy to generalize our model to regulation by **repressor TFs**, a case we explore in Supplementary Note 2.”